

# The effect of carbon monoxide on meiotic maturation of porcine oocytes

David Němeček[1], Eva Chmelikova[1], Jaroslav Petr[2], Tomas Kott[2] and Markéta Sedmíková[1]

[1] Department of Veterinary Sciences, Czech University of Life Sciences, Prague, Czech Republic
[2] Institute of Animal Science, Uhřiněves, Czech Republic

## ABSTRACT

Oxidative stress impairs the correct course of meiotic maturation, and it is known that the oocytes are exposed to increased oxidative stress during meiotic maturation in in vitro conditions. Thus, reduction of oxidative stress can lead to improved quality of cultured oocytes. The gasotransmitter carbon monoxide (CO) has a cytoprotective effect in somatic cells. The CO is produced in cells by the enzyme heme oxygenase (HO) and the heme oxygenase/carbon monoxide (HO/CO) pathway has been shown to have an antioxidant effect in somatic cells. It has not yet been investigated whether the CO has an antioxidant effect in oocytes as well. We assessed the level of expression of HO mRNA, using reverse transcription polymerase chain reaction. The HO protein localization was evaluated by the immunocytochemical method. The influence of CO or HO inhibition on meiotic maturation was evaluated in oocytes cultured in a culture medium containing CO donor (CORM-2 or CORM-A1) or HO inhibitor Zn-protoporphyrin IX (Zn-PP IX). Detection of reactive oxygen species (ROS) was performed using the oxidant-sensing probe 2′,7′-dichlorodihydrofluorescein diacetate. We demonstrated the expression of mRNA and proteins of both HO isoforms in porcine oocytes during meiotic maturation. The inhibition of HO enzymes by Zn-PP IX did not affect meiotic maturation. CO delivered by CORM-2 or CORM-A1 donors led to a reduction in the level of ROS in the oocytes during meiotic maturation. However, exogenously delivered CO also inhibited meiotic maturation, especially at higher concentrations. In summary, the CO signaling molecule has antioxidant properties in porcine oocytes and may also be involved in the regulation of meiotic maturation.

# INTRODUCTION

Carbon monoxide (CO) is an endogenously produced signaling molecule that affects several cellular functions. CO is endogenously produced by enzyme heme oxygenase (HO), which catalysis the reduction of heme to biliverdin, trivalent iron, and CO. HO is known in two isoforms (HO-1 and HO-2) that differ in expression and inducibility. HO-1 is an inducible isoform, and its expression increases after exposure to stressors. HO-2 is a constitutive isoform with basal expression (*Maines, 1997*; *Wu & Wang, 2005*; *Ryter & Choi, 2016*). CO produced by HO enzymes or delivered by CO donors regulates

Corresponding author
David Němeček,
nemecekd@af.czu.cz

cytoprotection, the cell cycle, metabolism, and cellular homeostasis (*Ryter, Alam & Choi, 2006*; *Kolluru et al., 2017*). It has been proved that the heme oxygenase/carbon monoxide (HO/CO) pathway is also involved in the regulation of reproduction. Zenclussen et al. examined the importance of HO-1 in mouse ovaries, and they have shown that HO-1 deficiency in mice reduces oocyte fertilizability (*Zenclussen et al., 2012*). In our previous publication, we demonstrated that exogenously delivered CO reduced caspase-3 activity and apoptosis in aged porcine oocytes (*Němeček et al., 2017*). However, it has not yet been investigated whether CO regulates the level of oxidative stress in oocytes.

Proper meiotic maturation in in vitro conditions is crucial for the development of biotechnological methods and the use of assisted reproduction methods. A balanced redox state is important for proper oocyte development (*Soto-Herasa & Paramio, 2020*), but in vitro cultivation systems of oocytes are characterized by increased oxidative stress (*Khazaei & Aghaz, 2017*).

Oxidative stress leads to deterioration of oocyte quality. This can disrupt meiotic maturation and eventually arrested cell cycle and activates apoptosis. Oocytes impaired by oxidative stress have lower fertilization and developmental potential (*Agarwal et al., 2012*; *Prasad et al., 2016*; *Wang et al., 2017*; *Xie et al., 2018*). Antioxidant supplementation of the culture medium can be used to avoid the harmful effect of oxidative stress. For example, the non-enzymatic antioxidant melatonin has been shown to improve oocyte maturation, fertilization rate, and rate of blastocyst formation (*Tamura et al., 2020*; *Soto-Herasa & Paramio, 2020*). The use of antioxidants can therefore ameliorate in vitro cultivation conditions, but novel substances are still needed to reduce oxidative stress during in vitro oocyte cultivation to improve oocyte quality.

Carbon monoxide has already been shown to reduce oxidative stress also in somatic cells (*Piantadosi, 2008*). Additionally, several factors that control meiotic maturation are simultaneously the cellular target of CO in somatic cells. These factors include MPF, MAPK, JNK2, p38 kinase, and potassium ion channels. Changes in the activity of these factors lead to the alteration of meiotic maturation (*Weston & Davis, 2007*; *Oh, Han & Conti, 2010*; *Huang et al., 2011*; *Miyagaki et al., 2014*; *Carvacho et al., 2018*). It has already been shown that CO regulates these signaling pathways in somatic cells (*Kim, Ryter & Choi, 2006*; *Peers et al., 2015*; *Ryter & Choi, 2016*), but studies on the effect of CO on oocyte meiotic maturation have been lacking. In somatic cells, CO reduces the expression of cyclins and thereby regulates the cell cycle (*Bauer et al., 2016*); by activating MKK3/p38 MAPK and NF-ƙB pathways, CO reduces Fas/Fas ligand interaction, increases expression of anti-apoptotic genes and thus prevents apoptosis (*Ryter, Ma & Choi, 2018*; *Kim & Choi, 2018*).

For these reasons, we assumed that CO could reduce oxidative stress in oocytes and thus decrease the effect of negative factors on in vitro meiotic maturation. Therefore, CO could increase the quality of in vitro matured oocytes. We focused on the HO/CO signaling pathway in porcine oocytes during meiotic maturation. Our aim was to determine the effect of the HO/CO signaling pathway on porcine oocytes and levels of ROS during their meiotic maturation.

## MATERIALS AND METHODS

### Porcine oocytes in vitro culture

Porcine ovaries were obtained from slaughtered prepubertal gilts (Large White × Landrace hybrids, slaughter weight 110 kg) that were in an unknown phase of the estrous cycle. Follicular fluid was obtained by aspiration of follicles (2–5 mm in diameter) using a syringe with a 20G needle. Only oocytes with intact cytoplasm and compact cumulus were used for further experiments. Oocytes were cultured in a 4-well multidish (Nunc, Roskilde, Denmark) in modified TCM-199 culture medium (Sigma–Aldrich, St. Louis, MO, USA), containing sodium bicarbonate (32.5 mM; Sigma–Aldrich, St. Louis, MO, USA), calcium L-lactate (2.75 mM; Sigma–Aldrich, St. Louis, MO, USA), sodium pyruvate (0.25 mg/ml; Sigma–Aldrich, St. Louis, MO, USA), gentamicin (0.025 mg/ml; Sigma–Aldrich, St. Louis, MO, USA), HEPES (6.3 mM; Sigma–Aldrich, St. Louis, MO, USA), 10% (v/v) foetal calf serum (Gibco BRL; Life Technologies, Darmstadt, Germany), and 13.5 IU eCG: 6.6 IU hCG/ml (P.G. 600; Intervet, Boxmeer, Netherlands). Oocytes were cultured to the stage of first (MI) or second meiotic metaphase (MII) for 24 or 48 h, respectively, in 1 ml of modified TCM-199 culture medium (5% $CO_2$, 39 °C).

The influence of CO on meiotic maturation was evaluated in oocytes cultured in a culture medium containing CO donors. We used CO donors CORM-2 (tricarbonyl dichlororuthenium (II) dimer; Sigma–Aldrich, St. Louis, MO, USA) and CORM-A1 (sodium boranocarbonate; Sigma–Aldrich, St. Louis, MO, USA). These donors differ in the kinetics of CO release: CORM-2 is a rapid CO releaser and CORM-A1 is a slow CO releaser. CORM-2, at concentrations of 5.0, 25.0, 50.0, and 100.0 μM dissolved in dimethyl sulfoxide (DMSO), or CORM-A1, at 25.0, 50.0 and 100.0 μM dissolved in distilled water, were used. The control group of oocytes was cultured in a culture medium containing inactive CORM-2 (ruthenium (III) chloride; iCORM-2; Sigma–Aldrich, St. Louis, MO, USA) at concentrations of 100.0 μM dissolved in DMSO or inactive CORM-A1 (iCORM-A1). iCORM-A1 was obtained by dissolving CORM-A1 (100.0 μM) in 0.1 M HCl, dissociating CO, and then neutralizing to pH 7.4. HO inhibitor Zn-protoporphyrin IX (Zn-PP IX; Sigma–Aldrich, St. Louis, MO, USA), at concentrations of 1.0, 2.5, 5.0, 10.0 and 25.0 μM dissolved in DMSO, was used to evaluate the effect of HO inhibition on meiotic maturation. The control group was cultured in a culture medium containing only DMSO at the final concentration of 0.25%. Cultivation in iCORM-2, iCORM-A1, and DSMO did not significantly affect meiotic maturation (Table S1).

### Reverse transcription polymerase chain reaction

The presence and amount of HO-1 mRNA and HO-2 mRNA were studied by reverse transcription polymerase chain reaction (RT-PCR). RNA obtained from oocytes was transcribed into cDNA with a High Capacity cDNA Achieve Kit (Applied Biosystems, Foster City, CA, USA); the final amount was 100 μl. Based on the knowledge of HO-1 and HO-2 sequences, the specific primers to amplify products were designed (Table S2).

Standard TaqMan PCR kit protocol was used (Applied Biosystems, Foster City, CA, USA) for RT-PCR. The reaction ran in a 10 μl reaction mixture, containing 500 nM of

gene-specific primers, 200 nM TaqMan MGB probe, 5μl Fast-TaqMan Universal Master Mix (Applied Biosystems, Foster City, CA, USA), and 1 μl cDNA and nuclease-free water. 7500 Fast Real-Time PCR System (Life Technologies, Carlsbad, CA, USA) was used for the reaction. Based on the obtained data, the relative amount of mRNA for each isoform was calculated using the $2^{-\Delta\Delta CT}$ arithmetic equation, according to the Ct method and expressed in comparison to GAPDH as an endogenous control.

### Immunocytochemistry

After completion of the oocyte cultivation period, zona pellucida was removed from oocytes with 0.1% pronase in phosphate-buffered saline (PBS) solution, and oocytes were fixed in 2.5% paraformaldehyde in PBS. After membrane permeabilization (0.5% Triton X in PBS with 0.01% bovine serum albumin; BSA), the oocytes were rinsed in 0.1% Tween 20 in PBS. Incubation with mouse primary monoclonal antibody (anti-heme oxygenase-1 or anti-heme oxygenase-2; Abnova; Taiwan; 1:200) was performed overnight (14–16 h) in 0.1% BSA and 0.1% Tween 20 in PBS at 4 °C. Oocytes were rinsed three times (0.1% Tween 20 in PBS) and incubated in secondary anti-mouse IgG antibody conjugated with fluorescein-5-isothiocyanate (FITC; Sigma–Aldrich, St. Louis, MO, USA; 1:100) at room temperature in 0.1% BSA and 0.1% Tween 20 in PBS for 1 h. The specificity of the primary antibodies was confirmed in our previous work by Western blot (*Němeček et al., 2017*). Chromatin was stained with 4′,6-diamidine-2-phenylindole (DAPI; Sigma–Aldrich, St. Louis, MO, USA). To control for secondary antibody non-specific binding detection, a control group of oocytes was cultured in a cultivation medium without the primary antibody. Oocytes were assessed using a confocal scanning microscope (Zeiss, Jena, Germany), and intracellular localization was determined based on signal intensity using NIS Elements 3.4 image analysis (Nikon, Tokyo, Japan). Data were expressed relatively as the mean signal intensity of the FITC fluorescence related to the basal signal intensity of the appropriate control group.

### Assessment of meiotic maturation of oocytes

After a culture period, cumulus cells were denuded by repeated pipetting through a thin-walled glass pipette. Then the oocytes were fixed for at least 24 h in a solution of ethanol and acetic acid (3:1, v/v), stained with orcein, and evaluated under a phase-contrast microscope. The stages of meiotic maturation were assessed based on nuclear maturation, as oocytes at germinal vesicle stage (GV; oocytes with visible germinal vesicle), metaphase I (MI; oocytes with chromosomes arranged in metaphase figure), and metaphase II (MII; oocytes with the extruded first polar body). Abnormal oocytes were evaluated as degenerated (Deg).

### Reactive oxygen species assessment

Production of reactive oxygen species (ROS) was measured in oocytes after 48 h of culture. After the cultivation period, oocytes were stained with 10 μM 2′,7′-dichlorodihydrofluorescine diacetate (Sigma–Aldrich, St. Louis, MO, USA) for 20 min at 39 °C. ROS production in oocytes was evaluated using a confocal scanning microscope (Zeiss, Jena, Germany).
Images were analyzed using NIS Elements. Data were expressed relatively as mean signal intensity related to the signal intensity of the appropriate control group.

## Experimental design

### HO-1 and HO-2 mRNA and proteins detection

The aim was to assess the expression of HO-1 and HO-2 mRNA and proteins in porcine oocytes during meiotic maturation. Detection of mRNA and proteins was performed using RT-PCR or immunolocalization, respectively in oocytes at GV, MI, and MII stages. For RT-PCR, each meiotic stage included 50 oocytes in six independent experiments. For the immunocytochemical method, localization of each isoform was performed on 25 oocytes for each meiotic stage in three independent experiments.

### The effect of HO inhibition

The aim was to assess the effect of HO inhibition on meiotic maturation. Nuclear maturation was evaluated after 48 h of oocyte cultivation in a culture medium containing Zn-PP. Each concentration of Zn-PP IX included 80 oocytes in three independent experiments.

### The effect of CO donor

The purpose was to determine the effect of CO donor on meiotic maturation of porcine oocytes. As in the previous experiment, nuclear maturation was evaluated after 48 h of oocyte cultivation in a culture medium containing CORM-2 or CORM-A1 donors. Each concentration of CORM-2 or CORM-A included 80 oocytes in three independent experiments.

### The effect of CO donor on ROS production

The aim was to assess the effect of CO donor on the production of ROS in porcine oocytes during meiotic maturation. Due to the similar effect of CORM-2 and CORM-A1 found in the previous experiment, only CORM-2 was used. The amount of ROS was measured by immunolocalization in oocytes cultured for 48 h in a culture medium containing CORM-2. Each concentration of CORM-2 included 30 oocytes in three independent experiments.

## Statistical data analysis

The data are presented as the mean ± SEM of at least three independent experiments. The data were statistically evaluated in the STATISTICA 12 program (Statsoft, Tulsa, OK, USA). Statistically significant differences between groups were assessed by analysis of variance (ANOVA) and multiple comparisons using Scheffé's method. A $P$-value of less than 0.05 was considered to be statistically significant.

# RESULTS

## HO-1 and HO-2 mRNA and proteins were detected in porcine oocytes

We have detected HO-1 and HO-2 mRNA in porcine oocytes during meiotic maturation by RT-PCR. The difference in HO-1 and HO-2 mRNA levels between oocyte categories (GV, MI, and MII) was not statistically significant (Table 1).

**Table 1 Expression levels of HO-1 and HO-2 mRNA and protein in porcine oocytes during meiotic maturation.** The expression level of mRNA was analyzed by RT-PCR in oocytes at GV, MI, and MII stages. The relative mRNA level was normalized to GAPDH and relative to oocytes at the GV stage. The amount of mRNA was calculated using the arithmetic equation $2^{-\Delta\Delta CT}$ according to the Ct method. The data are presented as mean ± SEM relatively to mean mRNA level in oocytes at the GV stage. The expression of HO-1 and HO-2 proteins is assessed by the immunocytochemical method in oocytes at the GV, MI, and MII stage. The date is expressed as mean ± SEM relative to the mean signal intensity of HO-1 or HO-2 in oocytes at the GV stage. The amount of mRNA and HO-1 and HO-2 proteins between different meiotic maturation stages were not significant.

| Culture period | mRNA | | Proteins | |
|---|---|---|---|---|
| | HO-1 | HO-2 | HO-1 | HO-2 |
| 0 h (GV) | 1.00 | 1.00 | 1.00 | 1.00 |
| 24 h (MI) | 0.61 ± 0.131 | 0.85 ± 0.086 | 1.05 ± 0.039 | 0.85 ± 0.030 |
| 48 h (MII) | 1.31 ± 0.308 | 0.98 ± 0.073 | 1.10 ± 0.028 | 0.94 ± 0.031 |

We evaluated the expression of HO-1 and HO-2 proteins in porcine oocytes during meiotic maturation (GV, MI, and MII). Using the immunocytochemical method, we demonstrated the presence of both HO isoforms at all stages of meiotic maturation (GV, MI, MII) (Fig. 1). The localization of the HO-1 isoform was dependent on the stage of meiotic maturation. In oocytes at the GV stage, HO-1 was localized primarily in the germinal vesicle; in the case of oocytes at MI and MII stage, HO-1 was detected in the perichromosomal region. The expression level of HO-1 was significantly lower in the cytoplasmic region than in the perichromosomal area (Fig. 1). In the case of the HO-2 isoform, we did not find a significant difference in the level of expression between the cytoplasmic and perichromosomal regions in oocytes at MI and MII stages. In oocytes at the GV stage, HO-2 expression was significantly lower in the germinal vesicle than in the cytoplasmic region. We did not find significant changes in the overall level of expression of HO-1 and HO-2 during meiotic maturation (Fig. 2).

## Inhibition of HO enzymes did not affect meiotic maturation

We evaluated the effect of HO inhibition on meiotic maturation by assessing nuclear maturation in oocytes cultured in a culture medium containing HO inhibitor Zn-PP IX. The cultivation of oocytes in a culture medium containing Zn-PP IX did not significantly affect meiotic maturation, and the proportion of matured oocytes did not differ between the control and experimental groups (99.2 ± 0.5% vs 96.4 ± 2.3%–100.0 ± 0.0% for control and Zn-PP-IX, respectively). The inhibitor slightly affected nuclear maturation only at the concentration of 5.0 μM, reducing the proportion of oocytes matured to the MII stage by 4.2%, as compared to the control group (99.2 ± 0.5% vs 95.0 ± 1.8 for control and Zn-PP-IX, respectively) (Fig. 3).

## Carbon monoxide inhibits the meiotic maturation of porcine oocytes

By assessing the effect of a CO donor on oocyte nuclear maturation, we found that the CO donor inhibited the nuclear maturation of cultured oocytes. Cultivation in a culture medium containing CORM-2 or CORM-A1 decreases the proportion of oocytes matured

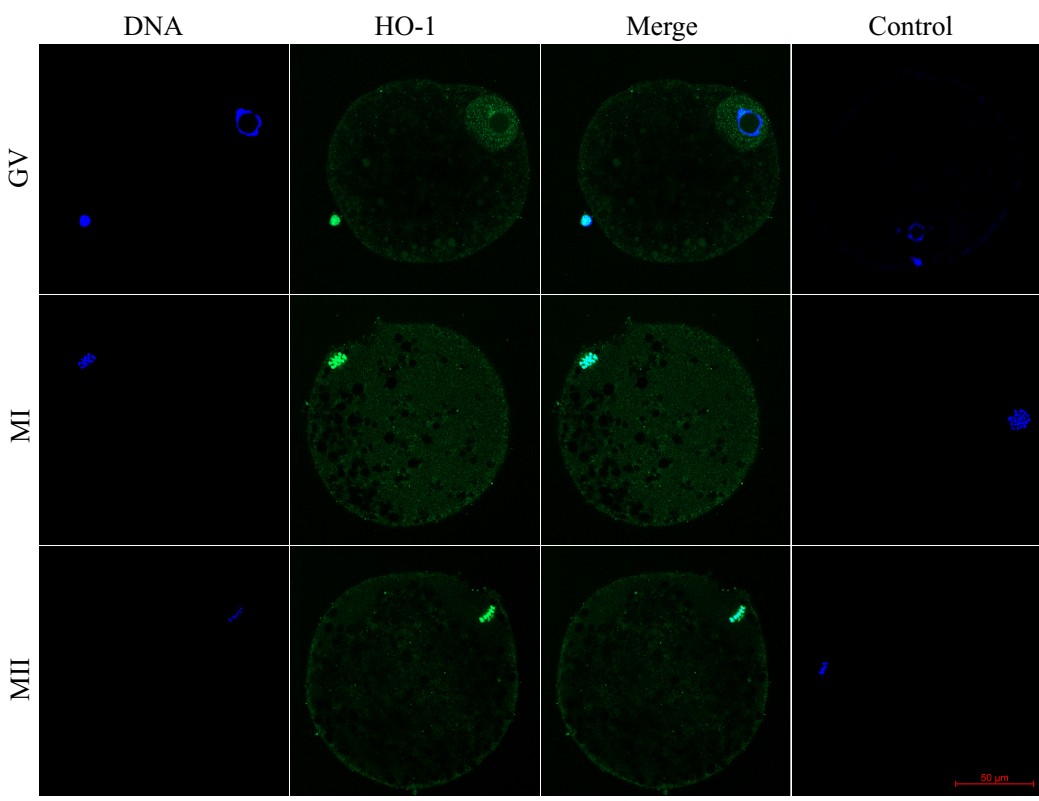

**Figure 1 Localization of heme oxygenase-1 (HO-1) in porcine oocytes during meiotic maturation.**
HO-1 is shown in green (FITC), chromatin is shown in blue (DAPI), magni-fied 400×; GV, germinal vesicle; MI, metaphase I; MII, metaphase II.

to the MII stage. After the oocytes cultivation in a culture medium containing CORM-2 or CORM-A1, CO donors inhibited nuclear maturation at all used concentrations. Meiotic maturation in media containing CORM-2 resulted in a significant reduction of the proportion of oocytes matured to MII ($85.5 \pm 1.1\%$ vs $55.1 \pm 1.9$–$67.9 \pm 1.4\%$ for control and CORM-2, respectively). CORM-2 at the concentration of 100.0 µM had the most pronounced effect on the proportion of oocytes matured to the MII stage. After cultivation in a culture medium containing CORM-2, CO arrested meiotic maturation at the MI stage and significantly increased the proportion of oocytes at the MI stage ($7.8 \pm 0.7\%$ vs $16.1 \pm 2.0\%$–$36.5 \pm 2.9$ for control and CORM-2, respectively). The effect was dose-dependent. CORM-2 at a concentration of 100.0 µM had the most potent inhibitory effect on meiotic maturation (most significant increase of the proportion of oocytes at the MI stage) (Fig. 4A).

Just like in the case of CORM-2, cultivation in a medium containing CORM-A1 resulted in a significant reduction of the proportion of oocytes matured to the MII stage ($82.0 \pm 0.5\%$ vs $45.5 \pm 2.5\%$–$61.4 \pm 2.5\%$ for control and CORM-A1, respectively). Also, CORM-A1 at the concentration of 100.0 µM had the most pronounced effect on the proportion of oocytes matured to the MII stage. Meiotic maturation was arrested mainly in the MI stage ($6.9 \pm 1.1\%$ vs $16.5 \pm 1.3\%$–$29.9 \pm 1.9\%$ for control and CORM-A1, respectively). Furthermore, oocyte cultivation in a culture medium containing CORM-A1

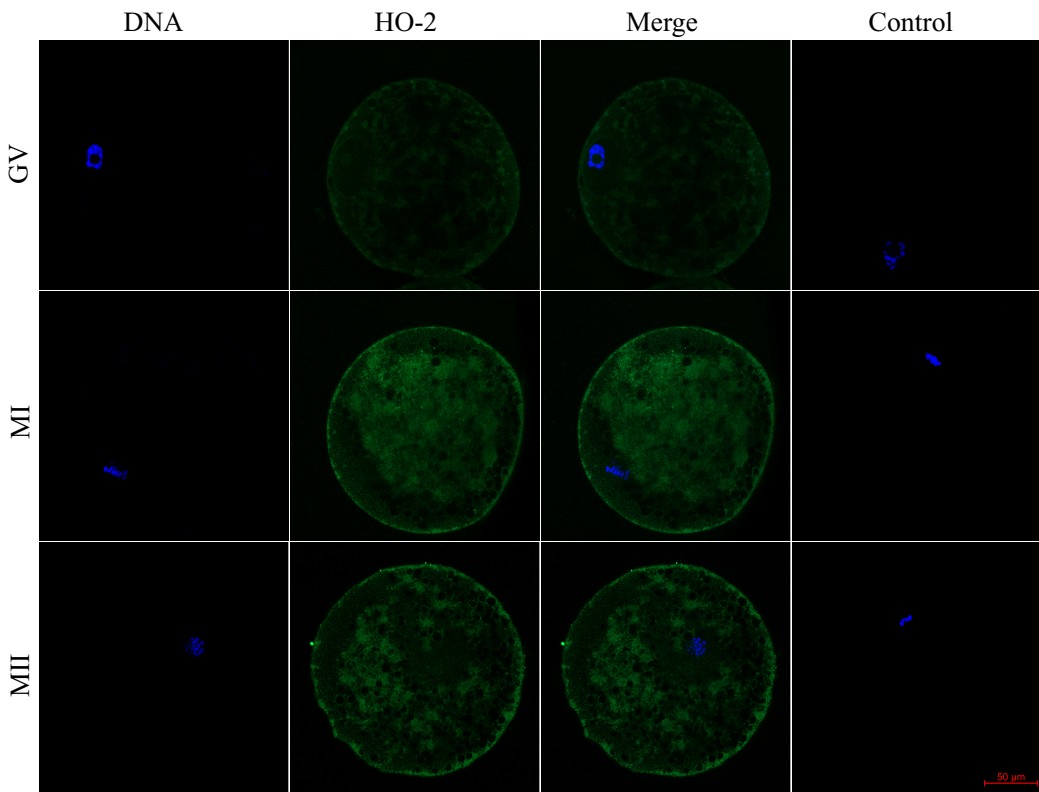

**Figure 2 Localization of heme oxygenase-2 (HO-2) in porcine oocytes during meiotic maturation.** HO-2 is shown in green (FITC), chromatin is shown in blue (DAPI), magni-fied 400×; GV, germinal vesicle; MI, metaphase I; MII, metaphase II.

at the concentration of 25.0 μM also increased the proportion of oocytes at the GV stage (5.6 ± 1.4% vs 14.5 ± 1.3% for control and CORM-A1, respectively) (Fig. 4B).

After analysis of oocytes cultured for 72 h in a culture medium containing CORM-2, we found that the oocytes did not complete meiotic maturation to the MII stage (80.5 ± 0.7% vs 42.4 ± 2.1%–58.2 ± 1.4% for control and CORM-2, respectively) and meiotic maturation remained arrested mainly at the MI stage (5.8 ± 1.0% vs 24.5 ± 2.0%–40.5% ± 1.9 for control and CORM-2, respectively). The COMR-2 had the most significant effect at the concentrations of 50.0 μM and 100.0 μM (Fig. 5).

## Carbon monoxide reduces the production of reactive oxygen species in porcine oocytes during meiotic maturation

We found that after meiotic maturation in a culture medium containing CO donor CORM-2, the amount of ROS in porcine oocytes decreased (Fig. 6).

Oocytes cultivated for 48 h in a culture medium containing CORM-2 led to a significant decrease in ROS production. All used CORM-2 concentrations had a significant effect on ROS level reduction. After meiotic maturation in a culture medium containing CORM-2 at 5.0, 25.0, 50.0, or 100.0 μM, the amount of ROS was reduced by 29.1%, 47.0%, 46.3%, and 48.9%, respectively. The effect of CORM-2 at the concentrations 25.0, 50.0, and 100.0 μM was not significant (Fig. 7).

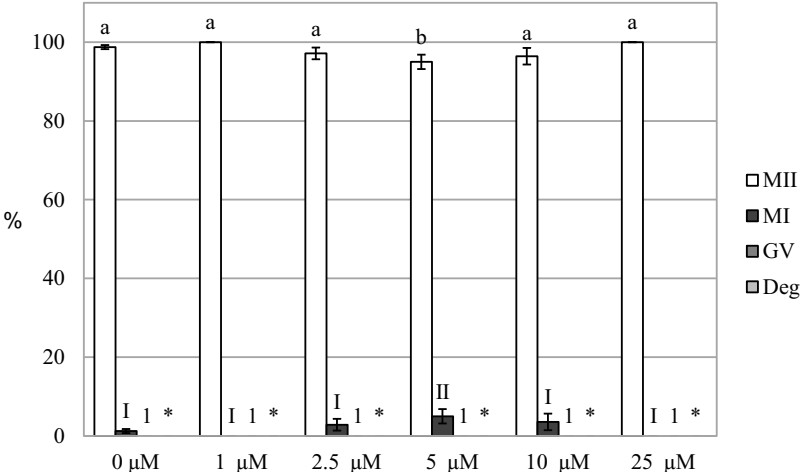

**Figure 3 Effect of HO inhibitor ZnPP-IX on the meiotic maturation of porcine oocytes.** The control group was cultured in a culture medium containing DMSO. Experimental groups were cultured in a culture medium containing ZnPP-IX at concentrations of 1, 2.5, 5, 10, or 25 μM. The date is expressed as mean ± SEM. Stages of nuclear maturation were evaluated as degenerate (Deg), germinal vesicle (GV), metaphase I (MI), and metaphase II (MII). Lowercase letters a and b show a statistically significant difference in the proportion of oocytes at the MII stage ($P < 0.05$). The numerals I and II show a statistically significant difference in the proportion of oocytes at the MI stage ($P < 0.05$). The number 1 shows a statistically significant difference in the proportion of oocytes at the GV stage ($P < 0.05$). The asterisk (*) shows a statistically significant difference in the proportion of Deg oocytes ($P < 0.05$).

## DISCUSSION

In the present work, we studied the importance of HO/CO for the meiotic maturation of porcine oocytes. Both heme oxygenase isoforms catalyze the oxidative degradation of heme. An excess of the heme molecule in the cell causes oxidative stress, and its removal by HO activity is important for cell protection (*Chiabrando et al., 2014*). During ovulation, the amount of heme in the ovaries increases, and it is believed that the enzyme HO protects the ovarian cells from the degradation of the heme molecule (*Zenclussen et al., 2012*). Degradation of heme by HO enzymes produces CO, a molecule with two faces. At high concentrations, CO has several toxic properties that are well known. However, at low concentrations, CO is an important signaling molecule that has cytoprotective, antiapoptotic, and antioxidative properties (*Ryter & Choi, 2016*; *Kolluru et al., 2017*). Considering the effects of HO/CO in somatic cells, we assume that CO could contribute to the protection of oocytes during meiotic maturation, especially in in vitro conditions. Our previous work demonstrated the presence of HO in oocytes exposed to in vitro aging (*Němeček, Dvořáková & Sedmíková, 2017*). In the present study, we evaluated the HO expression during meiotic maturation. We have not only proved the occurrence of both HO mRNA and proteins in porcine oocytes during their meiotic maturation, but we have also shown that CO has antioxidative properties in porcine oocytes. However, CO impairs meiotic maturation, particularly at high concentrations.

Cellular localization of the HO-1 isoform in oocytes predominated in the perichromosomal region, both in oocytes at the GV stage and in oocytes at the MI or MII

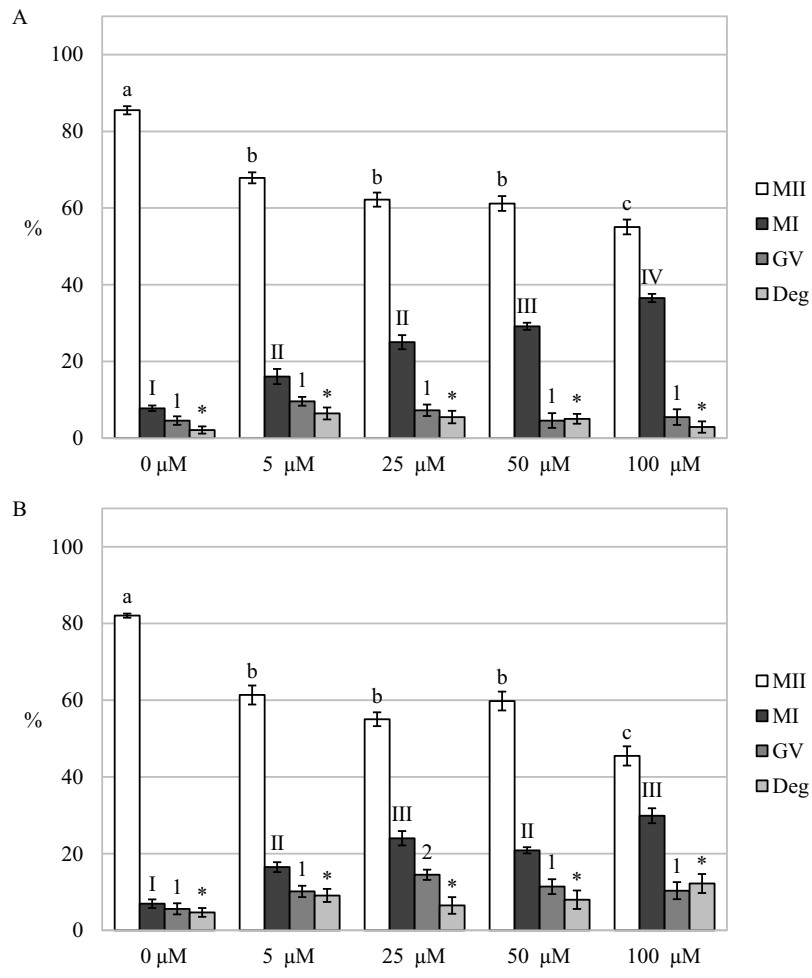

**Figure 4 Effect of CORM-2 (A) and CORM-A1 (B) on meiotic maturation of porcine oocytes.** The control group was cultured in a culture medium containing an inactive form of CORM (iCORM-2 or iCORM-A1). Experimental groups were cultured in a culture medium containing CORM-2 or CORM-A1 at concentrations of 5, 25, 50, and 100 µM. The date is expressed as mean ± SEM. Stages of nuclear maturation were evaluated as degenerate (Deg), germinal vesicle (GV), metaphase I (MI), and metaphase II (MII). Lowercase letters a, b and c show a statistically significant difference in the proportion of oocytes at the MII stage ($P < 0.05$). The numerals I, II and III shows a statistically significant difference in the proportion of oocytes at the MI stage ($P < 0.05$). The numbers 1 and 2 show a statistically significant difference in the proportion of oocytes at the GV stage ($P < 0.05$). The asterisk (*) shows a statistically significant difference in the proportion of Deg oocytes ($P < 0.05$).



stage. Localization of HO-1 was prevalent in the perichromosomal area also in aged porcine oocytes (*Němeček et al., 2017*) and in the nucleus of bovine granulosa cells (*Wang et al., 2018*). In somatic cells, HO-1 was mainly localized in the endoplasmic reticulum (*Dennery, 2014*), and translocation of HO-1 to the nuclear region may occur in response to stress factors (*Lin et al., 2007*). In the nuclear area, HO-1 regulates the activity of transcription factors such as AP-1 and NrF2, which increase the resistance of cells to oxidative stress (*Li Volti et al., 2004*; *Lin et al., 2007*; *Biswas et al., 2014*; *Tibullo et al., 2016*). It is shown, for example, that nuclear HO-1 regulates the expression of antioxidative

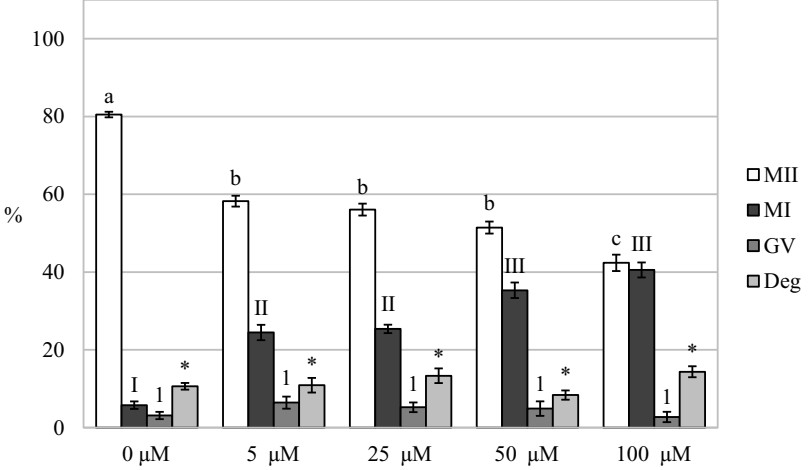

**Figure 5 Effect of CORM-2 on prolonged cultivation of porcine oocytes.** The control group was cultured in a culture medium containing an inactive form of CORM (iCORM-2). Experimental groups were cultured in a culture medium containing CORM-2 at concentrations of 5, 25, 50, and 100 µM. The date is expressed as mean ± SEM. Stages of nuclear maturation were evaluated as degenerate (Deg), germinal vesicle (GV), metaphase I (MI), and metaphase II (MII). Lowercase letters a, b and c shows a statistically significant difference in the proportion of oocytes at the MII stage ($P < 0.05$). Numerals I, II and III show a statistically significant difference in the proportion of oocytes at the MI stage ($P < 0.05$). The number 1 shows a statistically significant difference in the proportion of oocytes at the GV stage ($P < 0.05$). The asterisk (*) shows a statistically significant difference in the proportion of Deg oocytes ($P < 0.05$).

enzymes γ-glutamylcysteine synthetase, glutathione peroxidase, catalase, and methionine sulfoxide reductase. These enzymes are essential for resistance against oxidative stress (*Collinson et al., 2010*), and they are involved in the regulation of oxidative stress also in oocytes (*Cetica et al., 2001*). In addition to oxidative stress regulation in somatic cells, the HO/CO system also influences the cell cycle. Nuclear HO-1 may regulate the initiation of meiotic maturation through activation of the transcription factor Nrf2. It is already known that nuclear HO-1 regulates the activity of transcription factors (Nrf2 and AP-1) in somatic cells (*Li Volti et al., 2004*; *Lin et al., 2007*; *Biswas et al., 2014*; *Tibullo et al., 2016*). *Qiu & Yao (2017)* suggest that Nrf2 is involved in the initiation of meiosis since the inhibition of Nrf2 results in altered expression of the cell cycle-related genes and delayed progression in leptotene. For these reasons, nuclear HO-1 may regulate the activity of transcription factors and, thereby, meiotic maturation. In contrast to HO-1, the HO-2 isoform predominated in the cytoplasmic region. In somatic cells, HO-2 occurs mainly in the cytoplasm as a membrane protein of the endoplasmic reticulum (*Ma et al., 2004*; *Linnenbaum et al., 2012*). The HO-2 isoform is a constitutively active enzyme that does not respond to activation by stress factors. Also, in the porcine oocytes, we did not observe significant changes in HO-2 expression. It is believed that the HO-2 is responsible for the stable production of CO and thus can form a barrier against cellular damage, for example, by protecting against the negative effect of radicals derived from cellular metabolism (*Muñoz-Sánchez & Chánez-Cárdenas, 2014*). For these reasons, HO-2 could have a protective function also in oocytes. However, we have shown that inhibition of both HO isoforms does not significantly affect meiotic maturation. Though in the case of HO-1, it is

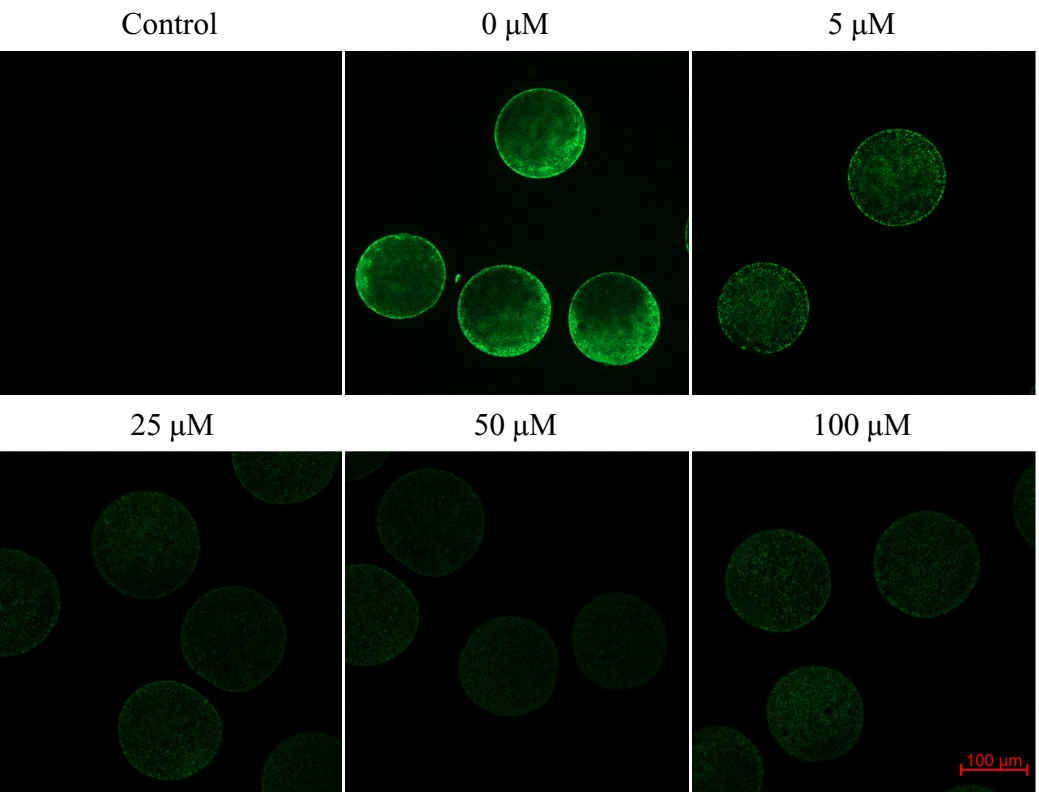

**Figure 6 Assessment of reactive oxygen species (ROS) production in the porcine oocyte during meiotic maturation.** The control group demonstrates the level of non-specific signal intensity in oocytes without treatment with 2,7-dichlorodihydrofluorescein diacetate method. The group 0 µM were cultivated in culture medium supplemented with iCORM-2. The experimental groups were a cultivated in culture medium containing CORM-2 at various concentrations. The level of ROS production was detected using the 2,7-dichlorodihydrofluorescein diacetate (green). Magnified 200×.

proven that both enzymatically active HO-1 protein and HO-1 protein with reduced enzymatic activity have antioxidative properties. This effect is probably due to the binding of inactive HO-1 to other proteins, such as transcription factors. Thus, HO-1 affects the transcription factors AP-1 and Nrf2 (*Lin et al., 2007*; *Dennery, 2014*) due to the localization of HO-1 in the perichromosomal region of porcine oocytes. For these reasons, HO-1 could affect the protein's activity despite the presence of an inhibitor. To assess the significance of HO during meiotic maturation, it would be useful, for example, to study the effect in animals that are deficient in HO enzymes. *Zenclussen et al. (2012)* examined the effect of HO-1 gene deficiency on oocyte ovulation, fertilization, and corpus luteum maintenance in mice. It is not known whether the HO-1 deficiency directly affected meiotic maturation, but the fertilization rate of oocytes from HO-1 deficient animals was decreased.

The gasotransmitter CO is already well known for its cytoprotective properties and also for the beneficial effect the exogenous delivery of CO can have on cells (*Wegiel, Chin & Otterbein, 2008*). Our results showed that cultivation in a culture medium containing CO donors led to a reduction in the level of ROS in porcine oocytes. At the same time,

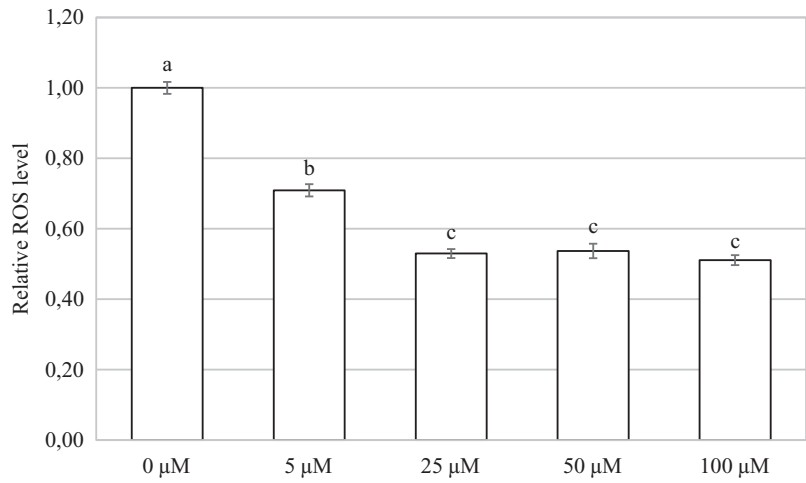

**Figure 7 Effect of CORM-2 on ROS level in porcine oocytes during meiotic maturation.** The control group was cultured in a culture medium containing an inactive form of CORM (iCORM-2). Experimental groups were cultured in a culture medium containing CORM-2 at concentrations of 5, 25, 50, and 100 μM. The level of ROS in porcine oocytes is expressed as the mean signal intensity of 2.7-dichlorodihydrofluorescindiacetate and is relative to the signal intensity in oocytes at the GV stage. The bars show the mean ± SEM. Lowercase letters a, b and c show a statistically significant difference in the level of ROS compared to the control group ($P < 0.05$).

however, CO arrested meiotic maturation, particularly at the MI stage. This effect was dose-dependent. It was shown in somatic cells that CO has an antioxidative effect and reduces the amount of ROS (*Pileggi et al., 2001*; *Brouard et al., 2002*; *Motterlini & Otterbein, 2010*; *Li et al., 2016*). CO is also involved in cell cycle regulation (*Wegiel, Chin & Otterbein, 2008*). In oocytes, ROS negatively affects meiotic maturation, fertilization, and developmental competence. Protection against oxidative stress is important, and cultivation in a culture media containing antioxidants improves meiotic maturation (*Combelles, Gupta & Agarwal, 2009*; *Prasad et al., 2016*). For these reasons, we consider the ability of CO to reduce the amount of ROS during meiotic maturation to be beneficial. The different concentrations did not have significantly different effects, and even the low concentration of CO led to a significant reduction in ROS levels. In aging oocytes, CO reduces caspase-3 activity and the occurrence of negative effects of postovulatory aging (*Němeček et al., 2017*). ROS are mainly responsible for the adverse effects of postovulatory oocyte aging; therefore, reducing the level of ROS may lead to improved oocyte quality and developmental potential (*Lord & John Aitken, 2013*; *Prasad et al., 2016*). The effect of CO on ROS levels has not yet been studied in aged oocytes, but because we proved the antioxidant effect of CO in oocytes, we assume that CO reduced the negative effects of aging by reducing the level of ROS. Thus, CO could have a beneficial effect during meiotic maturation and postovulatory aging of oocytes. However, ROS does not only have a negative effect on cells. It is also demonstrated that ROS is involved in the regulation of meiotic maturation and that a low level of ROS is essential for proper meiotic maturation (*Soto-Herasa & Paramio, 2020*). Oocyte cultivation in a medium containing high concentrations of ROS scavengers can lead to the inhibition of meiotic maturation (*Tiwari et al., 2016*). For example, supplementation of the culture medium with the non-enzymatic

antioxidants ascorbic acid and 3-tert-butyl-4-hydroxyanisole leads to the inhibition of meiotic maturation (*Khazaei & Aghaz, 2017*). In porcine oocytes, a high CO concentration could lead to a significant reduction of ROS level and alteration of the oxidative balance. Furthermore, a high concentration of CO can have a detrimental effect on meiotic maturation: meiotic maturation arrest. Therefore, we suggest that CO may be beneficial in low concentrations in in vitro oocyte culture.

The oocyte is not isolated in the cumulus-oocyte complex, but granulosa cells are important for its successful development. Granulosa cells supply nutrients and metabolites through gap junctions to oocytes and secrete paracrine signals to regulate oocytes. Oocyte also lacks several defense mechanisms that are provided by the granulosa cells. On the other hand, oocytes regulate granulosa cell proliferation and differentiation (*Alam & Miyano, 2020*; *Von Mengden, Klamt & Smitz, 2020*). Both HO isoforms are detected in granulosa cell (*Alexandreanu & Lawson, 2003*; *Bergandi et al., 2014*; *Wang et al., 2018*, *2019*). Particularly inducible HO-1 is involved in protecting granulosa cells against stressors. They are protected against oxidative stress via the Nrf2/HO-1 pathway (*Wang et al., 2018*). Increased stress leads to the induction of HO-1 expression and thus an increase in CO production in granulosa cells (*Wang et al., 2018*, *2019*). The level of stress may be too high, and the abilities of endogenous HO-1 may be overwhelmed. This then leads to cell damage. Significantly increased levels of HO-1 expression in granulosa cells may be associated with impaired oocyte competence (*Bergandi et al., 2014*), suggesting that an excessive ROS production can trigger oocyte damage (*Canosa et al., 2020*). The addition of protective agents, such as antioxidants, can then prevent cell damage. Antioxidant melatonin has been shown to enhance oocyte and embryo quality (*Rizzo, Raffone & Benedetto, 2010*). In granulosa cells, melatonin increases the expression of HO-1 and protects them from oxidative stress (*Yu et al., 2019*). Also, the induction of HO-1 activity by hemin increases antioxidant defenses and attenuates ROS generation and apoptosis in stressed granulosa cells (*Wang et al., 2019*). CO is mainly responsible for this protective effect. The addition of CO through the CORM-2 demonstrates that CO regulates the apoptosis of granulosa cells through the ERK1/2 pathway (*Wang et al., 2019*). We assume that CO may affect meiotic maturation also by affecting signaling pathways in granulosa cells. For example, the ERK1/2 pathway is important for the resumption of meiotic maturation (*Shimada, 2012*). On the other hand, CO is an activator of guanylyl cyclase. In the follicle cGMP is synthesized by guanylyl cyclase in granulosa cells and diffuses to the oocyte to inhibit the hydrolysis activity of PDE3A on cAMP, ultimately maintains the oocyte meiotic arrest (*Shuhaibar et al., 2015*; *Jaffe & Egbert, 2017*). The CO effect is probably complex and affects granulosa cells, oocytes, and their communication during meiotic maturation.

Meiotic maturation in in vitro conditions may result in oocytes with asynchronous nuclear and cytoplasmic maturation. This is important because adequate oocyte developmental competence requires synchronization between nuclear and cytoplasmic maturation. This condition decreases oocyte fertilizability and impairs early embryonic development (*Ali, Benkhalifa & Miron, 2006*; *Rybska et al., 2018*; *Leal et al., 2018*). Synchronization techniques based on the use of reversible meiotic inhibitors can prevent

this asynchrony. The purpose of this inhibition is to temporarily block meiotic progression during maturation. The block is then removed to allow the oocytes to mature under in vitro conditions (*Vanhoutte et al., 2009*; *Gil et al., 2017*; *Leal et al., 2018*). For example, temporarily arresting meiotic maturation by phosphodiesterase 3-inhibitor leads to an increase in oocyte quality and developmental potential (*Vanhoutte et al., 2009*; *Gil et al., 2017*; *Leal et al., 2018*). It would be an interesting question whether the application of CO to the culture system could lead to the synchronization of nuclear and cytoplasmic maturation. This could improve the quality of in vitro cultured oocytes.

Other signaling pathways may be responsible for the inhibition of meiotic maturation caused by CO, such as the JNK kinase-signaling pathway. In oocytes, the inhibition of JNK arrests meiotic maturation (*Huang et al., 2011*) also, it is shown that CO inhibits JNK activity in somatic cells (*Kim, Ryter & Choi, 2006*). CO can also regulate meiotic maturation through interaction with other gasotransmitters. It has already been shown that nitric oxide (NO) and hydrogen sulfide ($H_2S$) regulate meiotic maturation (*Bu et al., 2003*; *Nevoral et al., 2014*). In the case of $H_2S$, it has been shown to accelerate the meiotic maturation of oocytes (*Nevoral et al., 2014*, *2015*). CO inhibits the activity of $H_2S$ producing enzymes and thereby decreases $H_2S$ production (*Giuffrè & Vicente, 2018*). Inhibition of $H_2S$ producing enzymes by inhibitors leads to impaired meiotic maturation of porcine oocytes (*Nevoral et al., 2015*). In somatic cells, it has been shown that CO can reduce the level of cyclin proteins (*Bauer et al., 2016*); for example, MPF is an enzymatic complex composed of cyclin-dependent kinase 1 and cyclin B (*Kishimoto, 2018*). Furthermore, MPF activity is crucial for meiotic maturation, catalyzing entry into the M-phase of meiosis I and meiosis II (*Schmitt & Nebreda, 2002*). If CO affects cyclin B levels, then it could regulate MPF activity in this way.

## CONCLUSIONS

In summary, our work has shown that CO inhibits meiotic maturation and reduces ROS production in porcine oocytes. We assume that in oocytes, HO/CO may regulate the oxidative state and contributes to the protection against oxidative stress. Furthermore, we assume that CO probably affects some of the signaling pathways that regulate meiotic maturation. This leads to inhibition of meiotic maturation. To further assess the effect of CO during meiotic maturation, it is necessary to focus on the mechanism by which CO regulates meiotic maturation. We conclude that the HO/CO signaling pathway is an unexplored part that regulates meiotic maturation and that also regulates oxidative stress in oocytes.

### Funding

This work was supported by a grants—INTER-COST LTC 18059, CellFit COST Action CA16119 and by MZE-RO-0718. The funders had no role in study design, data collection and analysis, decision to publish, or preparation of the manuscript.

## Grant Disclosures

The following grant information was disclosed by the authors:
INTER-COST LTC: 18059.
CellFit COST Action: CA16119.
MZE-RO-0718.

## Competing Interests

The authors declare that they have no competing interests.

## Author Contributions

- David Němeček conceived and designed the experiments, performed the experiments, analyzed the data, prepared figures and/or tables, authored or reviewed drafts of the paper, and approved the final draft.
- Eva Chmelikova performed the experiments, analyzed the data, authored or reviewed drafts of the paper, and approved the final draft.
- Jaroslav Petr conceived and designed the experiments, performed the experiments, analyzed the data, authored or reviewed drafts of the paper, and approved the final draft.
- Tomas Kott performed the experiments, analyzed the data, authored or reviewed drafts of the paper, and approved the final draft.
- Markéta Sedmíková analyzed the data, authored or reviewed drafts of the paper, and approved the final draft.

## Data Availability

The raw data are available in the Supplemental Files.

## Supplemental Information

Supplemental information for this article can be found online at http://dx.doi.org/10.7717/peerj.10636#supplemental-information.

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
