# Peer review of "The effect of carbon monoxide on meiotic maturation of porcine oocytes"

_PeerJ, doi:10.7717/peerj.10636_

## Round 0.1 · original submission · Minor Revisions

The reviewers were incredibly positive about the manuscript and had only minor changes that need to be addressed. Please pay particular attention to reviewer's 1 and 2's comments to add recent references to the introduction and the discussion.

Reviewer 1 ·

Basic reporting

In general, it is a well written manuscript; however, the introduction section lacks recent references, they did not include articles from 2019 or 2020 I suggest including some recent articles with the use of antioxidants during in vitro culture.

Experimental design

Line 79, is it TCM-199 media instead of M199?
Provide the number of evaluated/used oocytes per experiment or total in the experimental design
Lines 91-92, why these concentrations were chosen?

Statistical analysis section
All figures only present the mean + SEM, not mean +- SEM, please modify

Discussion should include recent studies, 2019 and 2020 references are lacking

Validity of the findings

Findings are interesting; however, authors should indicate at which concentration CO inhibits maturation or reduces ROS. Conclusion is speculative and should only indicate the findings

·

Basic reporting

No comment.

Experimental design

No comment.

Validity of the findings

No comment.

Additional comments

1. The deduction that is given from line 63 is interesting, however, it would be important to point out if there is any previous study carried out in oocytes, even in other species, if not, mentioning the absence of other studies would give greater relevance to this work.
2. It would be important to check if there is more recent literature, in particular lines 54-57 and 66.
3. In lines 107-108, verify the meaning of RT-PCR, it would be "Reverse Transcriptase Polymerase Chain Reaction". Which would coincide with the transcription of RNA to cDNA that was carried out in the methodology.
4. On line 86, put CO2 with the subscript 2.

Overall, the work is very well done and promising. It leads to further studies.

Reviewer 3 ·

Basic reporting

no comment

Experimental design

no comment

Validity of the findings

no comment

Additional comments

Since the authors are stating that CO inhibits meiotic maturation, it would have been interesting to know whether or not this inhibition is reversible.
Since the authors are suggesting that CO may impact signaling pathways, it would have been interesting to know whether or not cumulus cels are contributing to this inhibition.

---

## Round 0.2 · Minor Revisions

The revised manusript is significantly improved. There are now just a few minor textual issues to address:

RT-PCR stands for reverse transcription PCR

Use American English (ferilization not fertilization)

Not sure why zona pellucida is italicized

Italicize in vitro

Signaling (one “l”) not “signalling”

---

## Round 0.3 · Minor Revisions

Please accept my apologies. The previous note autocorrected to leave out the "t" in fertilization. I just need you to correct to read "fertilizability" and "fertilization" instead of "fertilisability" and "fertilisation" and definitely not "ferilization"...

Please just revise with the correct spelling and the manuscript is otherwise acceptable for publication.

---

## Round 0.4 · accepted · Accept

Thank you for carefully addressing all the reviewers' comments and attending to the linguistic problems that I created for you. Congratulations on a nice paper.